# Volcano generated tsunami recorded in the near source

M. Ripepe [1] ✉ & G. Lacanna[1]

Volcano sector collapse and pyroclastic density currents are common phenomena on active volcanoes and potentially a fatal source of tsunami waves which constitute a serious hazard for local as well as distant coastal population. Several examples in recent history, warn us on the urgent need to improve our mitigation counter-actions when tsunamis have volcanic origin. However, instrumental record of tsunami generated by mass movement along a volcano flank are still rare and not well understood yet. Small tsunamis (≤1 m) induced by pyroclastic density currents associated to violent explosions of Stromboli volcano were recorded in near-source conditions (<1.6 km). We show how tsunami waveform remains unaltered regardless of the two orders of variability in the landslide volume and dynamics. This unprecedented record is also providing the lesson to develop unconventional warning strategies necessary when the tsunamigenic source is expected to be very close (<10 minutes) to densely populated coasts and with a limited time to issue an alert based on simulation of wave propagation and inundation.

Tsunami are mostly (~80%) triggered by the sudden displacement of the fault plane during large earthquake[1]. After the devastating tsunamis generated by the Sumatra (Indonesia) and Tohoku (Japan) earthquakes in 2004 and 2011, respectively, a lot has been done globally to improve our ability in predicting tsunami and in reducing the risk[2]. However, since 2011, many tsunamis have devasted the coastline worldwide claiming for more than 5000 victims[3]. Only a small percentage (10%) of all tsunamis are generated by aerial and submarine landslides and by the instability of volcanic flanks or volcanic activity[4]. In the historical record landslide-generated tsunamis have triggered local wave heights and runup as large as 100 m and 500 m[5], respectively, locally exceeding maximum wave and runup heights of tectonic tsunamis by more than an order of magnitude[6].

On 22 December 2018, the partial collapse of ~280 × 10⁶ m³ on the western flank of Anak Krakatau generated a 10–30 m tsunami wave on the closest (at ~3–5 km) islands of Sertung, Panjang and Rakata. Only 35–60 min after, waves of ~3 m high struck Sunda Strait in Indonesia in a 50 km range[7]. This event and a second tsunami in the same year in Palu Bay[3,8] eluded the warning system killing more than 2000 people. In the emblematic case of Anak Krakatau the rapid detection of the collapse combined with an efficient alert system on the coast could have prevented fatalities. Recently, the January 2022 violent eruption of Hunga volcano in Tonga has triggered worldwide atmospheric driven tsunami[9] which have globally surprised the modeling arriving almost 2 h before the expected "normal" earthquake-generated tsunami onset[10]. All these non-conventional tsunamis are calling for a better understanding of the tsunamigenic process and for a different approach in predicting and mitigating their effects.

The collapse of submerged flank of volcanic island are among the largest mass movements on Earth with potential volumes of order of km³ (10⁹ m³), such as those in the Hawaian[11], Canarian[12,13], Cape Verdean[14], Krakatau[15] and Stromboli[16] islands. These are considered one of the most dangerous geological phenomena able to trigger tsunamis propagating thousands of kilometres far from the source[17–19]. In addition, the collapse of part of the volcanic craters, or dome, and of the eruptive plume[20] is at the origin of pyroclastic hot gas and particles mixture density current which can run at velocities of ~200 km/h along the volcano slopes. The impact of the pyroclastic flow with the sea can originate tsunami with run-ups several meters high as observed during the Montserrat 1997 and 2003 eruptions[21] and the Rabaul 1994 eruption[22]. This extends the volcanic risk from the local to regional scale involving a large number of population and infrastructure near the coast of the volcanic island[14,15]. Most of the causalities associated to volcanic eruption at regional scale are in fact caused also by tsunami[1]

¹Dipartimento di Scienze della Terra, Università di Firenze, 50121 Florence, Italy. ✉e-mail: maurizio.ripepe@unifi.it

The dynamics of flank instability of volcanic islands are still poorly documented resulting in a great uncertainty on related tsunami generation. As many volcanic islands, Stromboli (southern Tyrrhenian sea, Italy) has in the north-western side a weak flank named Sciara del Fuoco (Fig. 1) which is the most impressive morphological feature of the volcano edifice with a mean slope, $\theta$, of $\sim 35°$ extending also below the sea level for a total length of almost 3000 m. The Sciara del Fuoco represents the subaerial part of a partially filled sector-collapse scar (Fig. 1). In the last 13 ka it has been the source of potentially tsunami-genic large-scale (in the order of $10^9$ m³) flank failures[23,24] generating tsunamis probably with run-up of ~50 m[18]. Recent work on paleo-events[16] has identified three well-preserved medieval (1300–1400 AD) tsunami deposits linked to the collapse of ~180 × $10^6$ m³ of the Sciara del Fuoco, comparable to the 2018 Anak Krakatau flank collapse, and making victims in the Neapolitan Gulf[16].

In the last century, volcanic activity at Stromboli has been responsible for at least six well-documented small-scale tsunamis[25,26] with the largest one on 30 December 2002 due to the partial collapse of 10–30 × $10^6$ m³ of the Sciara del Fuoco scar[27]. The tsunami badly damaged buildings at 10 m of elevation with inundation hundreds of meters in the nearby Stromboli coast, but also in the close (~20 km) islands (inlet in Fig. 1a). The tsunami was observed in several places along the coast of Italy, from the Campanian at north-east to the western part of Sicily southward[18].

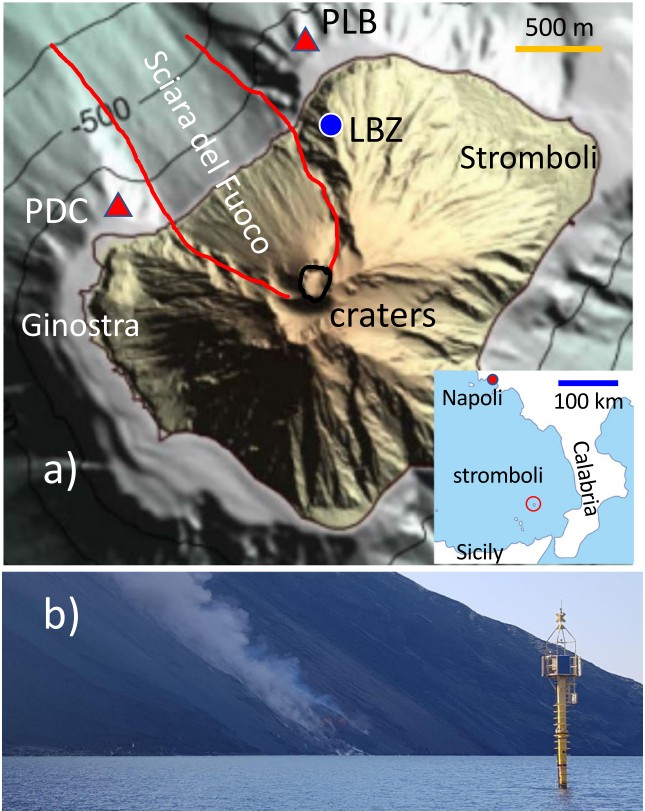

**Fig. 1 | Stromboli island and position of the sensors. a** PLB and PDC (red triangles) are the tsunami gauges deployed aside the Sciara del Fuoco slope which is the main source of tsunami. LBZ (blue circle) indicates the position of the visible camera. The red line is contouring the slope of the Sciara del Fuoco which extends for more than 1000 m below the sea surface. In the inlet position of Stromboli island in the Tyrrhenian sea. Calabria and Sicily are at a distance of ~55 km, while Naples is at ~235 km of distance. **b** Photo of PLB elastic beacon at ~300 m in front of the Sciara del Fuoco during the 9 October 2022 effusive eruption. The structure stands ~9 m above the sea surface and 24 m below the sea surface (see Supplementary Fig. 2).

Numerical simulations evidence that tsunami generated in the Sciara del Fuoco, will reach the populated coast of Stromboli in less than 3–4 min[18,27,28]. After only 20–30 min the whole Aeolian Arc and the coast of Calabria and Sicily (at ~50 km) would be impacted (inlet in Fig. 1). Waves would travel across the southern Tyrrhenian sea entering in the Neapolitan Gulf after 1 h and 20 min[28]. The short propagation time and the densely inhabited nearby coasts calls at Stromboli, as well as in many other volcanic islands, for a rapid detection system, able to issue an alert without human validation.

Two violent explosive eruptions (paroxysms) at Stromboli volcano produced in 2019 a few km-high (6–8 km) eruption columns, large tephra fallout ( ~$10^5$ m³) and pyroclastic density currents (e.g. 29), which propagated along the Sciara del Fuoco flank (Fig. 2). The impact of the density currents on the water generated moderate (meter-high) tsunamis recorded at a distance of <1.6 km from the source by permanent tsunami gauges (elastic beacons) and by the geophysical network operating at Stromboli (see Methods and Supplementary Note 1).

In this work, the dynamics of the pyroclastic flows derived by image analysis and the record of the tsunamis in near-source condition are used to constrain numerical models and to improve our ability to detect tsunamis of volcanic origin. We show how waveform and period of the tsunamis do not change with the landslide volume nor seems to be affected by landslide cinematics. Velocity and geometry of the mass movement along the volcanic slope do not significantly alter tsunami waveform and, as first approximation, the volume of the sliding body can be derived by the height of the tsunami using granular flow empirical approach. In case of tsunami generated near the coast, when the time to alert population is short (<10 min), this approximation can be used to derive almost in real-time a rapid assessment of the associated hazard. Combined with the automatic early detection of the tsunami provides the base for an efficient tool to mitigate the tsunami risk associated to large explosions and/or flank instability of volcanic islands.

## Results and discussion

In the summer 2019, on 3 July at 14:45:42 UTC and on 28 August at 10:17:15 UTC[29] two violent explosive paroxysms struck Stromboli island. The partial collapse of the 6–8 km high eruptive column[29] and, most probably, also of part of the crater rim, generated pyroclastic density current along the steep slope of Sciara del Fuoco impacting the sea surface and triggering tsunami waves (Fig. 3a).

On 3 July, georeferenced images of the visible camera located on the northern side of the Sciara del Fuoco at Punta Labronzo (LBZ in Fig. 1) show that during the initial phase of the paroxysm the fall out of large ejected blocks reached a distance of almost 500 m from the coast and ~1800 m from the vent, generating splash columns almost 50 m high (Fig. 2c and Supplementary Movie 1). The large quantity of ash and material ejected is masking the crater area and two density currents becomes visible only in the lower portion of the Sciara del Fuoco. The first density current propagates along the most southern part of the Sciara flank, covering the 730 m long visible portion of the slope in ~16 s with a front velocity $u_f = 45.6 m/s$ and entering in the sea at 14:46:10 UTC (Fig. 2c and blue star in Fig. 3a), only 28 s after the onset of the paroxysm. At 14:46:20 UTC. the second pyroclastic flow is clearly visible in the video (Supplementary Movie 1) entering in the sea only 10 s after the first one.

On 28 August, two fronts of another pyroclastic flow are visible in the more central part of the Sciara slope (Supplementary Fig. 1a) impacting the water at 10:17:49 UTC (Fig. 2f), ~34 s after the paroxysm onset. The time interval between the onset of the paroxysm and the impact with the water ( ~34 s) is ~6 s longer than on 3 July ( ~28 s). Image analysis shows that also this pyroclastic flow was moving at the constant velocity of $u_f = 45.7 m/s$ (Fig. 2g and Supplementary Fig. 1). The identical front velocity suggests that in both cases, density currents

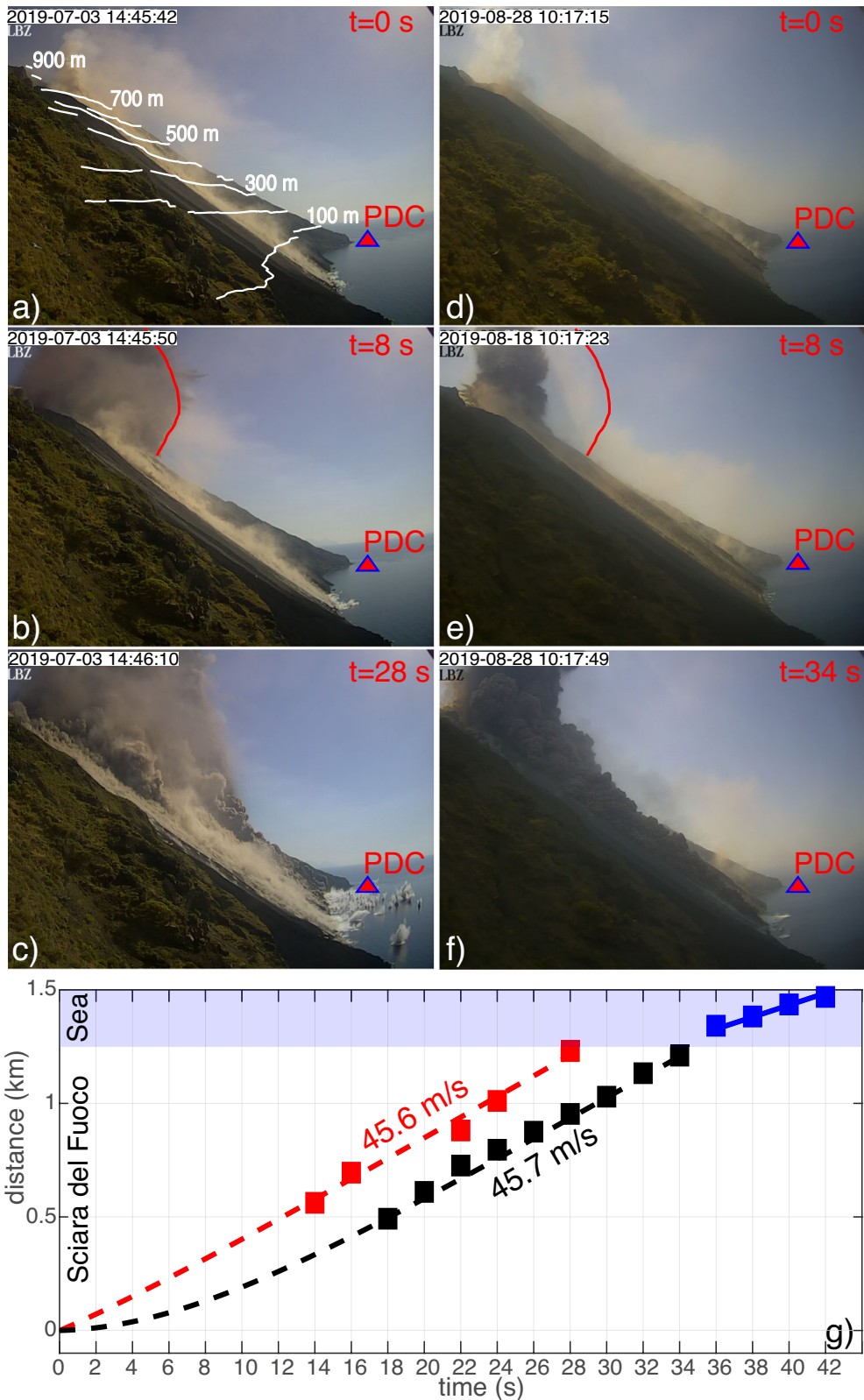

have almost reached a steady velocity already in the first 400 m of propagation, moving at a constant acceleration $du_f/dt = g\sin\theta$ (where $\theta = 35°$ is the mean slope of Sciara del Fuoco) after only 8 s. Considering that both density currents have travelled the same distance of 1250 m along the Sciara's slope with the same velocity, the time difference of 28 and 34 s, between the eruptive onset and the impact with the water, is indicating different initial conditions.

Considering a general multiphase mass flow landslide model[30], the analytical solution for the observed motion of the pyroclastic flows moving with the terminal velocity $u_f = 45.7 m/s$ along the Sciara del Fuoco can be calculated (see Methods) assuming a basal bed friction $\delta = 20°$[31] and a viscous drag coefficient $\beta = 0.0019$. The model indicates that terminal velocity is reached after only 15 s already in the first 400 m and it needs ~34 s to cover the 1250 m long Sciara del Fuoco

**Fig. 2 | Tracking pyroclastic density currents along the Sciara del Fuoco flank.** Snapshots of the videos taken by LBZ camera (Fig. 1) during the paroxysm occurred (**a**–**c**) on 3 July and (**d**–**f**) on 28 August 2019 show (**a**, **d**) the onset of two paroxysms, (**b** and **e**) 8 s after the onset and (**c** and **f**) the impact of the pyroclastic flow on the sea occurred (**c**) 28 s and (**f**) 34 s from the 3 July (snapshot is referring to the first frame after the impact) and the 28 August paroxysm onset, respectively. Images have been georeferenced (Supplementary Note 1) and the elevation contour map of the topography (white lines in Fig. 2a) has been overlapped on the volcano slope. The red triangle indicates the position of PDC elastic beacon which is at ~300 m from the coast. **c** During the 3 July 2019 pyroclastic flow, ~50 m high splash

produced by the impact of large blocks are visible offshore the Sciara del Fuoco at more than 500 m from the coast. The red line in **b** and **e** indicates the position of the explosive front during the 3 July paroxysm and evidences the large initial acceleration of the eruptive plume during the 3 July paroxysm. **g** Propagation of the pyroclastic front along the flank of the Sciara del Fuoco on the 3 July (red square) and 28 August (black square) events derived by the video taken at LBZ are well reproduced by the analytical model (see Methods) considering a terminal velocity of 45.7 m/s for both pyroclastic events and an initial velocity of 35 m/s in the case of the 3 July. The blue squares show the position of the pyroclastic front offshore the Sciara del Fuoco consistent with a propagation at 28 m/s on the sea surface.

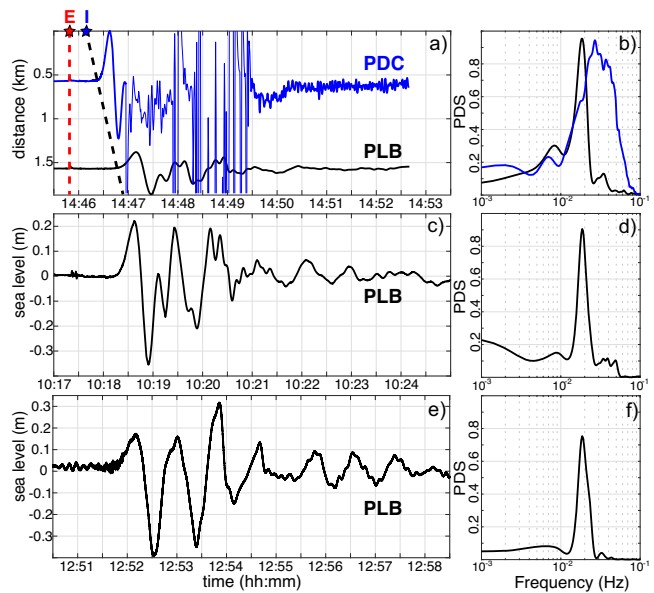

**Fig. 3 | Tsunami waves recorded at Stromboli.** **a** Tsunami recorded at PDC (blue line) and PLB (black line) gauges on 3 July 2019 and **b** their normalized power density spectra show the frequency shift from 0.025 Hz (40 s) at PDC to 0.018 Hz (55 s) at PLB, suggesting wave dispersion. Red star (E) indicates the onset of paroxysm at 14:45:42 UTC and the blue star marks the impact (I) of pyroclastic flow on the sea at 14:46:10 UTC derived by image analysis. The dashed black line is the travel time for a mean celerity of 39 m/s. **c** Tsunami recorded at PLB on 28 August 2019 and **d** normalized power spectrum. **e** The tsunami recorded at PLB on 19 May 2021 and **f** its normalized power spectrum. Note that the three tsunamis recorded at PLB have similar waveforms and the same frequency content at 0.018 Hz (55 s).

slope. While this is fully compatible with the movement measured for the 28 August pyroclastic flow (Fig. 2g), the analytical solution indicates that the 28 s measured during the 3 July event can be explained only assuming a 35 m/s initial velocity (Fig. 2g). The violent blast, also visible on the images was then accelerating the pyroclastic flow to its terminal velocity in only 8 s.

The observed density currents are composed by an upper dilute suspension and a darker basal concentrated granular avalanche with a total thickness of ~30 m, and a front almost ~200 m wide (Fig. 2f and Supplementary Fig. 1). Entering in the water, both pyroclastic flows in July and August 2019 drastically decelerate (Fig. 2c,f). While the dense basal part keeps flowing underwater, the ash-rich lighter and finer component of the density current runs on the sea surface at a mean velocity of 28.3 m/s (Fig. 2g and Supplementary Fig. 1d-f), quickly (in ~10 s) moving outside the camera field of view and propagating at least for 1 km from the shore line (Supplementary Fig. 1f). This suggests that only a small part of the visible total thickness moved underwater and is responsible for the tsunami.

For a sudden release of a finite volume of frictionless fluid down to an inclined plane, known as the dam-break problem, a gross estimate

of the front velocity is also given by $u_f \sim 1.4\sqrt{gH_c \cos\theta}$ [32,33], where $H_c$ is the height of the collapsed volume. In our case, the front velocity $u_f = 45.7 m/s$ would corresponds to an effective collapse height $H_c \sim 132 m$. According to laboratory experiments of a fluidized granular flow[33], we here consider that the minimum effective thickness ($h_f = 6.5\% H_c$) of the pyroclastic front responsible for the tsunami should be larger than ~9 m and smaller than 30 m.

### *Tsunami source time constrain*

After 43 s from the onset of the 3 July paroxysm, a tsunami wave with a peak-to-peak height $A_{PDC} = 2.59 m$ (Fig. 3a) and a period of 40 s (Fig. 3b) was recorded (at 14:46:25 UTC) first at the PDC gauge and after 26 s (69 s from the paroxysm onset) at the PLB gauge with an amplitude $A_{PLB} = 1.03 m$ (Fig. 3a) and a longer period of 55 s (black line in Fig. 3b). In line with video images (Fig. 2 and Supplementary Movie 1), differences in amplitude and time between the two elastic beacons indicate that the source of the tsunami was closer to the PDC than PLB gauge. The stretching of the period (Fig. 3b) from 40 s (at PDC) to 55 s (at PLB) is thus suggesting the dispersive nature of the tsunami in this near source conditions.

We used the arrival times at the two elastic beacons to search for the position of the tsunami source in an area 1600 m long by 500 m large extending offshore the Sciara del Fuoco (see Methods). The best solution gives a source located in the most southern-west part of the Sciara del Fuoco at a slant distance of ~610 and ~1590 m from the PDC and PLB gauges, respectively (Fig. 4), and a wave celerity c = 39 m/s. Source position does not coincide with the coast line but is ~150 m offshore (Fig. 4a) supporting the evidence that density current moved underwater and reached a depth of $H_o$ ~105 m below the sea level.

After ~30 s from the onset, the record of the tsunami at PDC (Fig. 3a) is contaminated by a very large amplitude and high frequency sequence of transients probably due to the elastic perturbations generated by the splash on the sea surface of the large blocks (Fig. 2c) and by the material transported during the flow. Part of the density current severely impacted on the elastic beacon PDC, at 300 m from the coast (Supplementary Movie 1), which for this reason was not operating during the tsunami on 28 August 2019. Notably, data transmission continues also during the pyroclastic flow suggesting that hot and dense ash is not shielding radio transmissions as expected. The record at PDC stops almost 8 min after the onset of the tsunami for damages on the radio link located on land due to the large fall out of incandescent lapilli and scoriae (Fig. 3a).

The tsunami on 28 August, is recorded at 10:18:20 UTC (65 s after the onset of the paroxysm) only by the PLB gauge and shows a positive onset with peak-to-peak amplitude $A_{PLB} = 0.6 m$ (Fig. 3c) smaller than what recorded on 3 July ($A_{PLB}$=1.03 m) but with the same period of 55 s (0.0182 Hz - Fig. 3d). Images provide a clear view of the pyroclastic density current touching the sea surface at 10:17:49 UTC (Fig. 2f) and fix the distance of PLB gauge at ~1170 m from the impact (Fig.4b). The time difference of 31 ± 1 s between the impact of the pyroclastic flow seen by video images and the tsunami onset gives an apparent celerity of c = 37.7 ± 1 m/s.

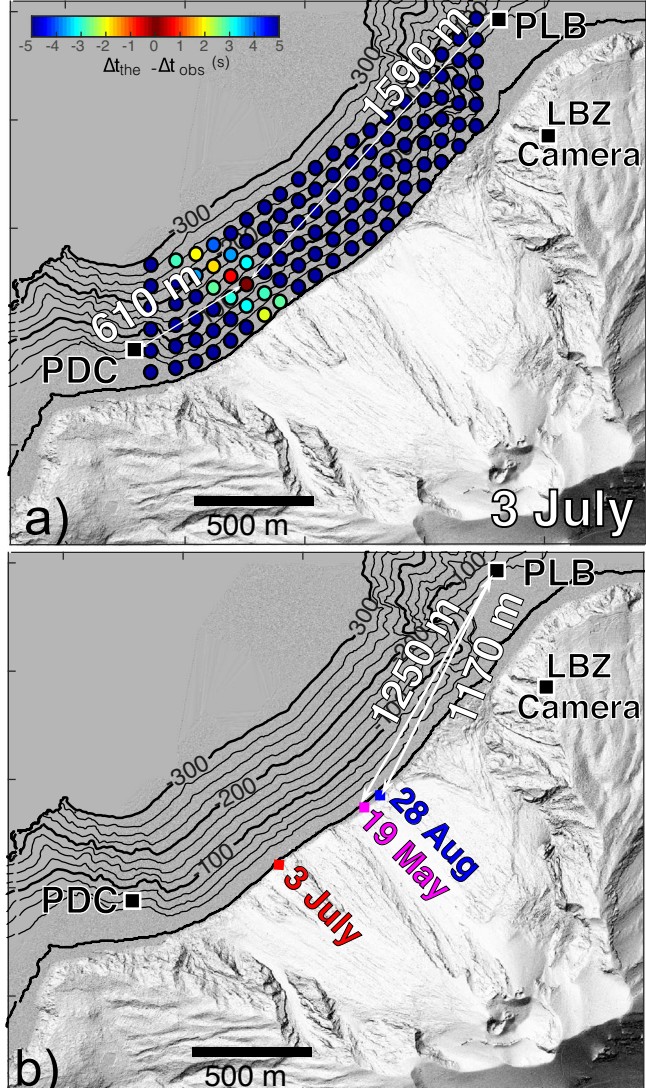

**Fig. 4 | Location of the tsunami source and impact of the pyroclastic flows. a** Bathymetry in front of the Sciara del Fuoco and the location of elastic beacons PDC, PLB and of the LBZ visible camera. Position of the source for the 3 July 2019 tsunami derived by comparing the observed and numerical arrival times at PDC and PLB gauges calculated using a finite time difference method (see Methods). **b** Location of the impact derived by image analysis (see Supplementary Information) are reported for the 3 July 2019 (red square), 28 August 2019 (blue square) and 19 May 2021 (magenta square).

## -Tsunami waveform characteristics

Differently from submarine slides, tsunami waves produced by sub-aerial landslides are characterized by a first positive onset[28,34]. The waveform similarity between the two tsunamis (Fig. 3a,c) recorded at PLB gauge, or the positive onset and the almost identical impact velocity ($u_f \sim 46$ m/s) of the density current, points to the same dynamics process. Considering the mean tsunami wave celerity $c = 39$ m/s, the period of 55 s recorded at PLB gives a characteristic wavelength $\lambda = 2145$ m which is larger than the maximum distance ($r = 1590$ m) between the impact area and the PLB elastic beacon (Fig. 4a) indicating that the two tsunamis were recorded in the very near-field condition ($r/\lambda \leq 1$).

Tsunami waveforms recorded at Stromboli are characterized by symmetrical wave profile with similar width and amplitude of the positive and negative pulse (Fig. 3a, c) typical of stokes waves[35] and consistent with the Froude number ($F_r = u_f / \sqrt{gH_0}$) of 1.43. In line with

the empirical relationship $F_r < (4 - 7.5S)$, the relative slide thickness $S = h_f/H_0$ is limited to 0.34 and provides the maximum slide thickness $h_f \leq 36$ m, fully consistent with the maximum thickness (30 m) of the pyroclastic front derived by the visible camera. These dimensionless quantities ($F_r = 1.43$ and $S \leq 0.34$) support the possibility to trigger tsunami with weakly nonlinear oscillatory wave[35]. Increasing the Froude number and/or for larger slide thickness, solitary wave (symmetrical wave with only a singular dominant crest) and bore wave (unsymmetrical wave both on the vertical and horizontal axes) could be generated[36].

The period of the 3 July tsunami recorded at the two elastic beacons (Fig. 3b) converts in a different wavelength $\lambda$ of 1560 and 2145 m at PDC and PLB, respectively. Experimental results indicate that in the near-field, at a dimensionless distance $2r/\lambda < 0.75$ from the splash zone[33], amplitude, $A$, of the tsunami is contaminated by the rapid vertical granular jet of water enriched by the air entrainment in the splash zone and this reflects the maximum elevation of the granular jet. In addition, laboratory experiments indicate that the amplitude of the wave during the propagation, up to $2r/\lambda \sim 2$, is independent on water depth $H_o$[33].

In this scenario, our recording stations, PDC and PLB, on 3 July are located at dimensionless distance ($2r/\lambda$) of 0.8 and 1.5, respectively, suggesting that both stations are well outside the splash zone and their amplitudes are not affected by the vertical granular jet of the flow. Tsunami waveforms recorded at PDC and PLB can be thus considered as representative for the leading wave in both 3 July and 28 August events.

## Comparing empirical and numerical models

Tsunamis generated by coastal landslides, or by pyroclastic density currents, are the result of the rapid transfer of momentum from the sliding mass to the water body during the impact and the penetration phases[37]. The resulting tsunami will propagate transversal along the coast and can drastically impact the near field regions by large wave runup[34].

Many numerical models have been developed to simulate tsunami waves generated by subaerial landslide[34] which have to account for a number of complex and quite often unknown parameters on the landslide dynamics, the interaction with the water and the bathymetric profile (e.g. angle of the slide, water depth, viscous drag coefficient, the speed and the duration of the sliding mass, thickness and width of the sliding front). Numerical models using both solid block and granular sliding body have been applied to simulate tsunami waves generated by the collapse of several millions of cubic meters of material along the Sciara del Fuoco slope[18,27,28,38]. Grounded on this extensive numerical modeling, we explore the possibility to use empirical equation to directly derive the volume of the pyroclastic flows from the height of the tsunami recorded during July and August, 2019 eruption.

Based on large scale two-dimensional laboratory experiments, several empirical relationships were derived to relate tsunami wave height to geometrical parameters of the sliding volumes[33,36–43]. Models to reproduce tsunamis generated by landslide and/or granular flow can be basically divided in two main groups: i) the release of solid block[39–41] and ii) the flow of a granular body on inclined plane[33,36–38,42,43]. The use of these empirical relationships is still debated and only few three-dimensional experiments are considering the lateral variation of the tsunami height respect to the direction of the sliding flow propagation.

We compared the solid block[41] and granular flow[37] empirical 3D models (see Methods) with previous results of the 3D non-hydrostatic NHWAVE numerical simulations of tsunami waves generated by aerial landslides occurring in the Sciara del Fuoco[28]. We assumed a truncated hyperbolic secant function[44] with a circular footprint to represent the volumes of the sliding block and the same initial parameters (Table 1)

**Table 1 | Parameters used to calculate the tsunami height and the landslide volumes of Fig. 5b**

| Tsunami | $h_f$(m) | $b$(m) | $u_f$(m/s) | $H_o$(m) | $r$(m) | $\gamma°$ | $A_g$(m) | $A_b$(m) | $A$(m) | $V$(m³) |
|---|---|---|---|---|---|---|---|---|---|---|
| NHWAVE | 30 | 670 | 70.0 | 150 | 1150 | 60 ± 4 | 7.3 | 10.8 | 7.20 | 4.7 × 10⁶ |
| NHWAVE | 45 | 670 | 70.0 | 150 | 1150 | 60 ± 4 | 11.1 | 14.5 | 11.50 | 7.1 × 10⁶ |
| NHWAVE | 74.7 | 670 | 70.0 | 150 | 1150 | 60 ± 4 | 16.3 | 21.3 | 17.70 | 11.8 × 10⁶ |
| 3 July 2019 | 20 | 175 ± 25 | 45.6 | 105 | 1590 | 65 | 1.03 | 2.9 | 1.03 | 2.08 × 10⁵ |
| 3 July 2019* | 20 | 175 ± 25 | 45.6 | 105 | 610 | 70 | 2.6 | 3.9 | 2.59 | 2.19 × 10⁵ |
| 28 Ago 2019 | 9.8 | 175 ± 25 | 45.7 | 105 | 1170 | 75 | 0.61 | 2.0 | 0.60 | 1.05 × 10⁵ |
| 19 May2021 | 6.5 | 175 ± 25 | 50.0 | 105 | 1250 | 75 | 0.55 | 1.5 | 0.54 | 0.71 × 10⁵ |

NHWAVE indicates the tsunamis simulated using the non-hydrostatic model where $A$ and $V$ are the tsunami amplitude and the slide volumes, respectively, calculated by Fornaciai et al.[28], $A_g$ and $A_b$ are the amplitude of the tsunami derived by the granular and solid block empirical equations, respectively, described in the Methods. For the 3 July 2019, 28 August 2019 and 19 May 2021, $A$ indicates the amplitude of the tsunami measured at the gauge PLB. The asterisk indicates the parameters used to calculate tsunami amplitude and landslide volume for the PDC tsunami gauge. $\gamma$ is the angle between the landslide direction and the recording station.

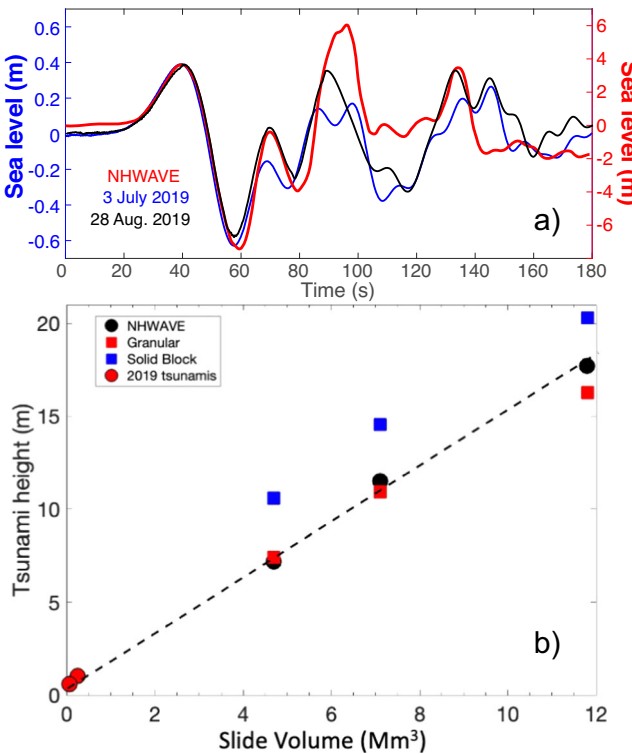

**Fig. 5 | Tsunami waveforms. a** Tsunami waveforms modeled[28] at PLB gauge by NHWAVE (red line) for an aerial slide of 7.1 × 10⁶ m³ compared with the observed tsunami recorded at PLB on 3 July (blue line) and 28 August 2019 (black line). The tsunami recorded on 28 August 2019 has been normalized to amplitude of the 3 July 2019 tsunami. **b** Analytical solutions for the solid block (blue squares) and granular flow (red square) empirical models (see Methods) using the same parameters to calculate tsunami waves with the 3D non-hydrostatic NHWAVE numerical modeling[28] for three different aerial landslide volumes (black circles). Granular flow model (see Methods) was also applied to calculate the volume ($V$) of the pyroclastic flows from the tsunami amplitude ($A$) occurred in 2019 (red circle). The linear fit using the NHWAVE solutions and 2019 granular flow modeling ($V = 6.8x10^5 \cdot A - 3.9x10^5$) is represented by dashed line. All the parameters used to derive the data presented in Fig. 5b are detailed in Table 1.

landslides remains constant among the different simulation[28], the height of the modeled tsunami is mainly function of the landslide maximum thickness ($h_f$) and shows a linear correlation with landslide volume (Fig. 5b).

A granular model is then applied to the 2019 tsunamis considering the circular footprint volume shape with width $150 < b < 200$ m. For the 3 July tsunami, the empirical model fits the wave height at both tsunami gauges ($A_{PDC} = 2.59$ and $A_{PLB} = 1.03$ m) for a slide thickness $h_f = 20 ± 4$ m and a volume of $2.14 ± 0.5 × 10^5$ m³. In the case of the 28 August tsunami height $A_{PLB} = 0.6$ m recorded at PLB (Fig. 2b) is compatible with a volume of $1.05 ± 0.21 × 10^5$ m³ and a slide thickness of $h_f = 10 ± 1$ m (Table 1).

These results are in good agreement with the linear relationship between tsunami amplitude and landslide volume found in the NHWAVE numerical simulations (Fig. 5b). Besides, the tsunamis simulated by NHWAVE numerical modeling (see Fig. 3f in ref. 28) have a waveform remarkably similar to the tsunamis recorded both on July and August 2019 (Fig. 5a). Surprisingly waveforms and period (T = 55 s) of the tsunamis remain the same regardless of the two orders of variability in the landslide volume (from -10⁷ to -10⁵ m³), the different location of the tsunami source and the landslide dynamics (Froude number). This similarity is evidencing the reliability of NHWAVE[46] numerical model and that in near-field conditions, different source geometry ($b$ and $h_f$), position of the impact ($r$ and $\gamma$) and Froude number ($u_f$ and $H_o$) of the landslide (see Table 1) do not affect tsunami waveform. This suggests that a linear relationship ($V = 6.8x10^5 \cdot A - 3.9x10^5$) between tsunami height ($A$) and landslide volume ($V$) can be considered reasonably acceptable, with implications on our ability to promptly assess the hazard along nearby coast.

## Tsunami and crater rim collapse

On 19 May 2021, the partial collapse of the crater rim induced by the increase of the internal conduit pressure associated to a small lava overflow triggered a dense flow of incandescent material which impacted the water at 12:51:15 UTC (http://lgs.geo.unifi.it/bulletins/ ? bulletin=171) generated a tsunami recorded at 12:51:49 UTC at the PLB gauge (Fig. 3e). At a distance of $1250m$ (Fig. 4b), the tsunami was $A_{PLB} = 0.54m$ height, almost the same of the 28 August 2019 tsunami, and it moved with a similar celerity $c = 36.7$ m/s. The dense ash cloud rising up from lapilli and scoriae ejected by the explosions made not possible to track the front velocity of the pyroclastic density current using our visible camera, we then assumed the mean front velocity $u_f = 50m/s$ previously estimated by thermal camera[47]

Using the same granular landslide empirical equations (see Methods) and considering a slide width $b$ ranging between 150 and 200 m, the amplitude of the tsunami is consistent with a volume of $0.71 ± 0.15 × 10^5$ m³ for a slide thickness $h_f = 6.5 ± 1$ m. This volume is well in harmony with the $0.8 × 10^5$ m³ volume of material collapsed

used in the numerical simulation for volumes ranging between 4.7 and $11.8 × 10^6$ m³. The empirical granular equations (see Methods) give at PLB gauge tsunami height which within ±3% nicely fit the NHWAVE numerical simulations[28] (Fig. 5b and Table 1). In line with previous conclusions[34,45], we found that the analytical solid block model is overestimating by a 20–50% the tsunami height of the numerical simulation (Fig. 5b and Table 1). Given the dynamics of the modeled

from the north flank of the NE crater estimated from images taken by helicopter immediately after the failure[47].

## Tsunami detection algorithm

In the last three decades, detection algorithms and early warning systems for tsunami generated by earthquake sources have been strongly improved. These algorithms generally recognize the tsunami if the sea level amplitude[48] or its first derivative[49] is exceeding a given threshold[50,51]. Tsunami detection is mainly used to validate the warning issued by seismic network[52], only after the source and the magnitude of the earthquake have been defined. This warning strategy is not very effective for tsunami generated by large mass sliding in the water, such as landslides and volcano flank instabilities[53].

The two-tsunamis that occurred at Stromboli in the 2019 summer provide the first record since the installation of the elastic beacon in 2008 and probably they represent the first record of tsunami generated by a volcano in near–source conditions. The maximum recorded amplitude, at 610 m from the coast (Fig. 3a), was of 2.59 m and fortunately the wave had no significant run up and then a negligible impact on the Stromboli's coast which during the summer are visited by more than 5000 people every day. However, these small events offered the unique possibility to test our ability to prompt and automatically deliver an alert to population.

Our algorithm is grounded on the short-term (STA)-long-term (LTA) average ratio method (see Methods and Supplementary Note 3) which is generally used in seismology[54] to automatically detect earthquakes. The algorithm tested using 5 years long data-set recorded both at PDC and PLB gauges guarantees to automatically alert if a tsunami as large as 40 cm will occur in the worst sea conditions and with no false alert (Supplementary Fig. 4).

The tsunami on 3 July 2019 was detected by the Early Warning algorithm at PDC at 14:46:32 UTC and at PLB at 14:47:07 after only 7s and 16 s, respectively, from the onset and before the maximum amplitude is reached (Fig. 6a, b; Supplementary Movie 1). The same performance is observed on 28 August 2019 when tsunami was automatically detected at 10:18:31 after 11 s from the onset even though this event has a positive amplitude of only 0.2 m (Fig. 6c), indicating the high sensitivity of the algorithm.

On 28 August the tsunami early warning alert was still being tested using the PLB gauge, but it allowed Civil defence authorities to activate the acoustic alert manually at 10:18:31, only 11 s after the onset of the tsunami (Fig. 6c) and less than 4 min before the tsunami reached the populated coast of Stromboli[28]. In our knowledge this is the first time an early warning is issued for a tsunami generated by a pyroclastic flow.

## Towards early warning for volcanic tsunami

The 2019 tsunamis at Stromboli represent, as far as we know, an unprecedented record of a volcano tsunami at its early stage, when it is still forming. Constrained by physical parameters such as the velocity and the geometry of the density current flow, our records give the unique opportunity to test empirical solutions based on solid block approximation[41] and granular materials[37] models. As expected[34,45], the granular material empirical solution better resolves the source parameters than the solid block model which overestimates the tsunami height (Fig. 5).

These results line up with the previous observed linear proportionality between the volume of the sliding body and the height of the tsunami wave (e.g. ref. 55). This is suggesting that at least for tsunami generated by the collapse of material sliding along the steep slope of the Sciara del Fuoco, when no physical parameters of the source are available, the volume of the sliding material can be derived as first approximation by an empirical linear relationship. Regardless of this very simplistic approach to derive volumes, this approximation allows to give in real time a rough estimation of the volumes of the body triggering the tsunami, which becomes of primary importance

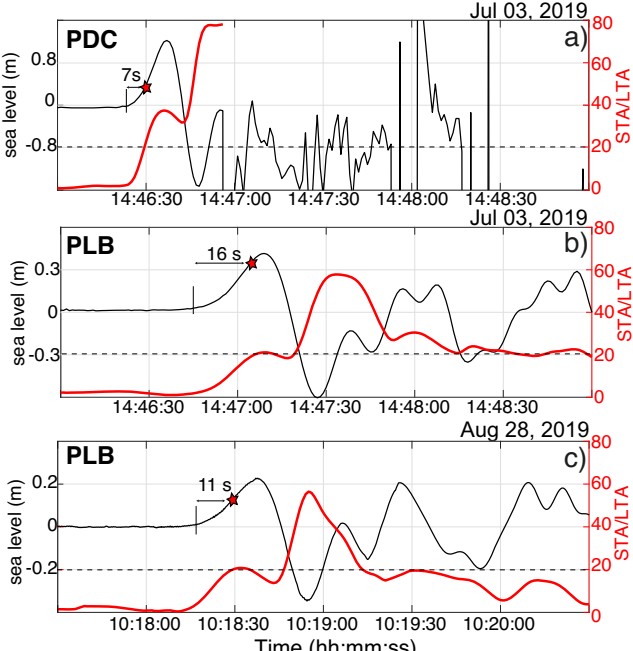

**Fig. 6 | Tsunami waves detected by STA/LTA method.** Tsunami waves (black lines) recorded **a** at PDC and **b** PLB gauges during the 3 July 2019 explosive paroxysm, the stars indicate the time when the STA/LTA ratio (red lines) is above the threshold (dashed lines) for the automatic detection. The time difference between the onset of the tsunami and the detection (STA/LTA ≥ 20) is 7s and 16 s for PDC (**a**) and PLB (**b**), respectively. **c** Tsunami wave (black line) recorded at PLB on the 28 August 2019. In this case the STA/LTA ratio (red line) is beyond the detection threshold (black dashed line) already 11 s (red star) after the onset.

when a prompt predictive numerical simulation for the inundation area along the coast is required. Inundation scenarios assuming different sliding volumes could be, in fact, pre-calculated and used to relate the amplitude of the tsunami detected by the gauges to the effects on the nearby coasts in almost real-time. Models will allow to define a minimum threshold in the tsunami height below which no alert should be delivered to the population when limited effects along the coast are expected. The approximation between tsunami amplitude and sliding volume will thus make the warning system not only fast in detecting tsunami (within seconds from the onset) but also more effective in the definition of the associated hazard.

We suggest that systems like the one developed at Stromboli could be used also in other scenarios namely when the source of the tsunami, including earthquakes and submarine landslides, is too close (within 10 min) to densely populated coast and with a limited time to generate simulation on wave propagation. The record of tsunami waves in the near-source is also shedding lights on the physical properties of the tsunami dynamics which would help to improve our understanding on this poorly monitored natural phenomena.

## Methods

### The elastic beacons tsunami gauge system

Stromboli volcano is monitored by University of Florence (LGS), National Institute of Geophysics and Volcanology (INGV) and University of Palermo, with an integrated network of several geophysical and geochemical sensors (including broadband seismic stations, infrasound network, ground deformation, SO2 cameras, multigas sensors and thermal as well visible cameras) specifically designed to provide timely information on the possible magma intrusion which can lead to the tsunamigenic instability of the Sciara del Fuoco flank. For this reason, two tsunami gauges (Fig. 1) were deployed by the Laboratorio di Geofisica Sperimentale, LGS, (http://lgs.geo.unifi.it/) of the

University of Florence in 2008 and 2017 offshore the Sciara del Fuoco, at 260 m and at 350 m distance from Punta dei Corvi (PDC) and Punta Labronzo (PLB) capes, respectively (Fig. 1). The extremely rough conditions of the sea in front of the Sciara del Fuoco (significative waves up to 8 m with periods of -12 s), called for using elastic beacons (Fig. 1) instead of floating buoys as infrastructure for measuring the sea level at Stromboli. Tsunami are measured at the seabed (46 and 50 m depth at PDC and PLB, respectively) by hydrostatic pressure sensors sampled at 125 Hz (Supplementary Fig. 2). At this depth, wave dispersion reduces by -87% the effect of the sea wave at periods <13 s and it preserves waves in the period range of 50–200 s (Supplementary Note 2, Supplementary Fig. 2b), typical of the tsunamis induced by landslide in general and volcanic activity in particular[38,56]. At depth larger than 1000 m, the attenuation is very large even at periods above 50 s (20% at 100 s, see Supplementary Fig. 2b). Sampling rate and sensors depth are the crucial factors to guarantee the best signal-to-noise ratio and for developing an efficient detection system for tsunami generated by volcanoes (Supplementary Note 2).

## Pyroclastic flow velocity

The velocity $u(x,t)$ of the pyroclastic flow along the Sciara del Fuoco slope has been calculated using the multiphase mass flow landslide model[30]:

$$\frac{\partial u}{\partial t} + u \frac{\partial u}{\partial x} = \alpha - \beta u^2 \tag{1}$$

where $\beta$ is the viscous drag coefficient and $\alpha$ is representing the net driving force in the system:

$$\alpha = g^x - g^z \left( \alpha_s \mu + h_g \right) \tag{2}$$

which depends on the component $g^x$ and $g^z$ of the gravity acceleration along ($x$) and perpendicular ($z$) to the slope, respectively, the volume fraction of the solid particles $\alpha_s = 0.56$[33], the basal friction coefficient ($\mu = \tan\delta$), where $\delta$ is the basal friction angle (20°), in the mixture material[31], and $h_g$ is approximating the surface gradient $\partial h_f/\partial x$ of the flow thickness $h_f$ along the slope. The time-independent steady-state motion $u(x)$ can be developed and takes the general solution[30] of:

$$u(x) = \left\{ \frac{\alpha}{\beta} \left[ 1 - \frac{\beta}{\alpha} u_o^2 \right] \frac{1}{\exp(2\beta(x - x_o))} \right\}^{1/2} \tag{3}$$

where $u_o$ is the initial velocity at the initial position $x_o$. For a sufficiently long distance and long time, the motion of the flow reaches a steady-state and Eq. (3) becomes $u = \sqrt{\alpha/\beta}$ which represents the terminal velocity of the flow. Assuming a terminal velocity of 45.7 m/s we thus calculate the viscous drag coefficient $\beta = 0.0019$.

## Tsunami source location

We applied finite difference time domain method based on a nonlinear shallow-water model of tsunami wave propagation[57] to calculate the travel times $t_{PLB}$ and $t_{PDC}$ needed to cover, the distances $r_{PDC}$ and $r_{PLB}$ between the two (PLB and PDC) elastic beacons and the 19 × 6 nodes equispaced every 100 m along the shoreline and offshore the Sciara del fuoco (Fig. 4a, Supplementary Movie 2). We used a gaussian source function 1500 m large with 1 m amplitude, to calculate the location of the tsunami source by comparing the observed delay time between the tsunami wave recorded at PLB and PDC ($\Delta t_{obs} = 26$ s) and the numerical delay time ($\Delta t_{the}$). The best agreement between the observed and theoretical delay time indicates an unique solution for a source -150 m offshore the Sciara del Fuoco and a celerity c = 39 m/s (Fig. 4).

## Landslide volume

The 3D landslide geometry used for slide volume estimation was a truncated secant function[44] having an elliptical foot-print on the slope, with length $b$ and width $w$ and vertical cross sections with maximum thickness $T$ varying according to hyperbolic secant functions:

$$\zeta = \frac{T}{1-\varepsilon} \left\{ \text{sech}(k_b \xi)\text{sech}(k_w \eta) - \varepsilon \right\} \tag{4}$$

with $k_b = 2C/b$; $k_w = 2C/w$; $C = \text{acosh}(1/\varepsilon)$ and with the truncation parameter $\varepsilon = 0.717$. For the specified $\varepsilon$, the slide volume is estimated by the formula $V = 0.3508bwT$[58].

## Solid block model

For a 3D block model experiment, the relative propagation time $t_s$ of the landslide underwater, after the splash, is the key parameter[41] to derive the tsunami source parameters[40]. This time can be empirically derived from the dimensionless surface of the landslide front impacting the water ($S_l = bh_f/H_0^2$), the Froude number ($F_r$) and the slope of the Sciara del Fuoco ($\theta = 35°$):

$$t_s = 0.43 S_l^{-0.27} F_r^{-0.66} (\sin\theta)^{-1.32} \tag{5}$$

where $b$ is the width and $h_f$ is the thickness of the block[41]. From the dimensionless time of propagation of the block underwater (Eq. (5)), we can then estimate the peak-to-peak maximum wave height, $A_B(r,\gamma)$ as function of the direction cosine ($\cos\gamma$) of the tsunami wave propagation and the relative distance $R = (r/H_0)$ from the source to the elastic beacons:

$$A_B(r,\gamma) = H_o \cdot 0.07 \left(\frac{t_s}{S_l}\right)^{-0.45} R^{-0.44} (\sin\theta)^{-0.88} \exp(0.6\cos\gamma) \tag{6}$$

Using a propagation angle $\gamma = 60° \pm 4°$ and a distance $r = 1150$ from the impact area of the different landslide simulated scenario[28], Eq. (6). predicts a maximum tsunami amplitude at the PLB gauge by 20–50% larger than expected (Fig. 5b and Table 1).

## Granular flow model

We use the 3D parametric equations developed by Mohammed and Fritz[37] to calculate the amplitude of the tsunami wave as function of the water depth $H_o$, radial propagation distance $r$ and angular direction $\gamma$ with respect to the landslide flow axis. Multi variable regression analysis leads to the empirical equations for the wave amplitude:

$$A = \left(k_a R^{n_a} + k_b R^{n_b}\right) H_o \cos\gamma \tag{7}$$

where the parameters $k_a$ and $n_a$ are relative to the first crest

$$\begin{aligned} k_a &= 0.31 F_r^{2.1} S^{0.6} \\ n_a &= -1.2 F_r^{0.25} S^{-0.02} B^{-0.33} \end{aligned} \tag{8}$$

whereas $k_b$ and $n_b$ are relative to the first wave trough

$$\begin{aligned} k_b &= 0.7 F_r^{0.96} S^{0.43} L^{-0.5} \\ n_b &= -1.6 F_r^{-0.41} L^{-0.14} B^{-0.02} \end{aligned} \tag{9}$$

defined as function of the Froude number ($F_r$), the relative distance $R = (r/H_0)$, the relative slide thickness $S = h_f/H_0$, relative slide width $B = b/H_0$ and relative slide length $L = V/(h_f bH_o)$. As for the Block Model[41], we used Eq. (7). to calculate the tsunami maximum height for the same landslide scenario calculated numerically by Fornaciai et al.[28]. We found that Granular Model of Eq. (7) is sensitive to the propagation direction ($\gamma$) but in a range between 56° and 64° better fits the results

of the numerical simulation (Fig. 5b) predicting the maximum wave height within the ±3% of error (Table 1).

## Tsunami early warning algorithm

The tsunami early-warning system developed for Stromboli is based on the short-term (STA) and long-term (LTA) average (Supplementary Note 3 and Supplementary Fig. 3). The STA and LTA values used by the Early Warning algorithm to detect tsunami waves has been specifically tuned for the sea wave conditions at Stromboli. Whereas STA is sensitive to rapid fluctuations in the sea amplitude, the LTA provides information on the signal background noise. We set the LTA window to 4500 s to include at least 300 times the longest sea wave period of 15 s (typical of Mediterranean sea), whereas the STA window was fixed to 40 s to get the highest ratio for tsunami with a period ranging between 50 and 200 s, as those expected for tsunami triggered by subaerial and underwater sliding mass like the one occurred at Stromboli in 2002[27,28,38] and at Anak Krakatau volcano[56]. To improve the signal-to-noise ratio, a signal decimation and a low-pass filter are applied before STA/LTA ratio is calculated (Supplementary Note 3). When the signal-to-noise ratio is high, the STA/LTA method is able to detect tsunami only few seconds after the onset, and several tens of seconds before the maximum amplitude is reached, providing the most as timely as possible alert (see Supplementary Note 3 for more details). The threshold ratio STA/LTA = 20 is 5 times larger than the ratio measured at Stromboli during the worst sea conditions (Supplementary Fig. 4) and gives the highest reliability to detect the tsunami before the first maximum amplitude is reached. The automatic alert is triggered when the STA/LTA ratio is larger than the detection threshold (>20) at both PDC and PLB stations for at least 120 s (Supplementary Fig. 3). This last logical filter increases the reliability of the system minimizing the possibility of false detections.

This tsunami detection algorithm is active since 9 September 2019, when the system was connected automatically to the syrens of the Italian Civil Defence alert system and it can work equally with two or only one tsunami gauge. No false alerts have been issued in this last four years. On 4 December 2022 for the first time an alert was automatically triggered by a tsunami 1.5 m (peak-to-peak) high induced by a pyroclastic flow originated by the partial collapse of the northern part of the crater sector (http://lgs.geo.unifi.it/bulletins/?bulletin=94592).

## Data availability

All data generated or analyzed during this study are included in this published article and its supplementary information file or available from the corresponding author upon reasonable request.

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

## Acknowledgements

This work was financed and strongly supported since 2004 by the Department of the Italian Civil Protection (DPC) in the framework of the DEVNET project. We acknowledge Resinex Trading s.r.l. for their constant technical support and to share with us their long-term experience on elastic beacons. The efficiency of the infrastructure has been maintained in the years by the constant care and dedicated passion for the sea of Stefano Maggioni by SteMag s.r.l. to which we are deeply grateful. Finally, we thank all the people that in the years have encouraged our efforts.

## Author contributions

M.R. conceived the technical solution of the tsunami monitoring system. G.L. has developed the algorithm at the base of the early-warning system and the tsunami source location. M.R. and G.L. wrote the manuscript and prepare the Figures.

## Competing interests

The authors declare no competing interests.
