## [Peer Review File · Nature Communications]

Volcano generated tsunami recorded in the near sourceREVIEWER COMMENTS

Reviewer #1 (Remarks to the Author):

Dear colleagues,

The data you present here is unique and highly valuable. The scientific community has very little data, especially in the near field, on the tsunamis generated by volcanic eruptions. It is a topical issue, following the events of Anak Krakatau in 2018 and Tonga in 2022. This work deserves to be widely disseminated and published in a prestigious journal such as Nature Communications.

However, there is still a big effort to be made on the manuscript. The methodological protocols are well detailed and the data are well presented and processed in a relevant way. But the structure of the paper is a bit chaotic and needs to be completely reworked. Several sections appearing in the results or discussion are in the area of methods or introduction. The separation between the results and their discussion is unclear (see my suggestions in an annotated version of the manuscript).

This is one of the reasons that leads me to recommend a major revision.

Here are also some other major points to improve:

- The literature on tsunamis generated by subaerial landslides is not well "digested", which is understandable because the authors have rather an expertise in volcanology, but this is felt in certain phrases (eg lines 263-266). This literature is certainly abundant, but it lacks a paragraph mentioning the landslide parameters that influence tsunami generation (eg from experimental and numerical studies).

- The notion of "near-field" must be defined. Many tsunami studies consider the near-field as an impacted area close to the source (typically several kilometers away), but here you are almost at the source. Your record is indeed unique, but two references should still be cited where measurements are available: Rabaul 1994 (tide-gage record of the tsunami generated by PDC: Nishimura et al 2005 In: K. Satake (ed.) Tsunamis: case studies and recent developments, Springer, 43-56) and Montserrat 2003 (dilatometer data: Mattioli et al 2007 GSA Bull 35). To come back to this idea of "near-field" I would suggest that either you keep writing "very near-field" everywhere in the manuscript, or you write "at its source" or "near-source".

- The discussion on existing alert systems is somewhat peremptory and involves simplifications. In any case, we cannot compare a system dedicated to ocean-wide tsunamis. For sources such as landslides and volcanic eruptions (except for atmospheric waves), local systems nested within regional systems must be used. This is, moreover, what is being done in Stromboli... but you are more aware of these steps than I am. This discussion should be redirected to the need to set up local warning systems and dedicated procedures on active volcanoes close to the coast, rather than to "criticize" the existing systems which remain essential for other types of sources.

- It's a shame not to have NHWAVE simulations with volumes comparable to the events of July and August 2019. This would bring real added value to your work. This is not a criticism, but a suggestion to consider.

See also other comments on the annotated pdf attached to this report.

I am impatient to see this work published, even if there are still some efforts to provide. Good luck.

Raphaël Paris

Reviewer #2 (Remarks to the Author):

The authors, Drs. Lacanna and Ripepe, present unprecedented records of tsunamis generated by volcano sector collapse and pyroclastic density current from Stromboli volcano. The authors first show near-field records of tsunamis caused by pyroclastic flows, for the first time. They also attempt to locate the tsunami source, and present a technique to tsunami alert system by detecting volcanic tsunamis from Stromboli volcano.

The near-field tsunami records and their analyses shown in this manuscript are scientifically important in the hot topic of volcanic tsunamis. Also, their analysis based on tsunami modeling with the pyroclastic flow modeling, and their tsunami detection system will be helpful for coastal communities around Stromboli volcano. I think that the research contents may be potential for publication from Nature Communication. However, I need to say that the manuscript, at this present form, is not well prepared and contains a lot of careless typos, duplicate words, grammatical errors, and wrong references to figures, most of which may be avoided by brief checks. Also, the structures need extensive revision; for example, research backgrounds which should be explained in Introduction appear in the discussion part in a redundant way. These issues, mainly related to their presentation, unfortunately make it difficult to understand the scientific importance of their research. In the following, I list up major and minor comments one by one. These issues would be required to resolved before further consideration.

[Major comments]

1. Paper structures

First of all, the structure is not well organized. Here I list up parts that may need to be revised.

- Lines 41–46: I do not think the Tonga eruption case needs to be explained in an independent paragraph, because this study's focus is rather on Anak Krakatau type. This part can be shortened and merged into the previous paragraph.

- Lines 85–105: This part does not describe results of this study and should be moved to the

introduction section.

- Three paragraphs in Discussion (Lines 257–261, 334–340, 392–398, and 420–428) do not fit here and most of the parts are just repeated explanations that have already appeared in earlier parts. I recommend that the authors revised the parts carefully to avoid the redundant repeating.

2. Estimation of initial tsunami location and tsunami propagation speed

In “Tsunami data wavefield”, the authors assumed straight ray paths at constant speeds. But clearly the tsunami wave speeds depend on the bathymetry, and the ray paths are not straight. Are the assumptions, straight ray paths and constant speeds, are truly valid? I think the ray path should travel slightly to a deeper part offshore and bent back to the shore due to the bathymetry slope. I recommend the authors perform basic tsunami simulations (for example, I guess linear long-wave simulation from a simple Gaussian-like source is enough) distributed in the same way, and compare the simulation and observed arrival times.

[Minor comments]

- Lines 27–32:

Please provide references for the following parts for the two sentences, “More than 10% of all tsunamis are generated by landslides or volcano collapses with subaerial, partially submerged or submarine origins” and “In recorded history landslide-generated tsunamis have triggered local wave heights and runup as large as 100 m and 500 m, respectively...”

- Line 52: extent => extends

- Line 54: cost => coast

- Line 70: The connection between this and the previous paragraphs are not clear. How about starting this paragraph by “In addition to the flank collapse, the impact of the pyroclastic flow with the sea can originate tsunami with run-ups several meters high, as observed during the Montserrat 1997 and 2003 eruptions (23) and the Rabaul 1994 eruption (24)”?

- Lines 76 and many other parts: The date representations are not unified; July 3rd, 3 July, or 3 July. Sometimes, 3th July.

- Line 78: which propagating => which propagates

- Line 98: “in in” => in

- Line 124: What does “involving most probably also part of the crater rim”?

- Line 133: What do you mean by “in the second portion of the Sciara del Fuoco”?

- Line 136: yellow square => yellow star

- Line 137: clear visible => clearly visible

- Line 159: during the 3 July => during the 3 July event

- Line 160: the its => its

- Line 164: concentrate granular avalanche => concentrated granular avalanche

- Line 165: almost ~200m large => almost ~200m high

- Sections “Tsunami data wavefield” and “Tsunami waveforms”: Please reconsider the titles of these sections. The present titles are similar, and readers may not understand the differences at first glances. Please describe what you’ve done in these sections by concise but informative titles.

- Line 182: After 43 seconds => After 43 s

- Line 186: amplitudes and time difference => differences in amplitude and time

- Lines 204 and 210: the second tsunami on August 28 => the tsunami on August 28 (this reads as two tsunami events occurred on August 28)
 - Lines 221–223: The waveform similarity between the two tsunamis (Fig. 3a,c) recorded at PLB gauge, or the positive onset and the almost identical impact velocity (~ 46 m/s) of the density current, points to the same dynamics process.
 - Line 305: remarkable similar => remarkably similar
 - Line 319: which impacting => which impacts
 - Line 329: $0.67 \pm 0.15 \times 10^5$ => $(0.67 \pm 0.15) \times 10^5$
 - Line 334: the two-tsunami occurred => the two tsunamis that occurred
 - Line 334: are => gives or provide
 - Line 347: the a priori => the a-priori
 - Line 348: a quite difficult => quite difficult
 - Line 351: wait => waiting
 - Line 350: implies to => implies
 - Line 355: LTA/STA method has been already attempted by a previous study for tsunami detection, for example, Wang et al. (2020).
 - Line 406: previous observed => previously observed
- Also, please provide references to the previous studies.
- Line 460: Fig. 4b => Fig. 4a
 - Lines 460: what does "travel times should equals the \diamond PLB ≤ 41 s and \diamond PDC ≤ 15 s time interval" mean?

Figure, tables, and the captions

- Line 694: extend => extends
- Figure 2's captions: multiple "3th July"
- Figure 3a: vertical axis "distancel" => distance
- Line 754: 28 August 2019 (red line) => 28 August 2019 (black line)
- Figure 5a: In legend, "Ago." => "Aug."
- Line 733: remove "always"
- Line 738: "different three different" => three different
- Line 759: indicates that => indicates
- Table 1: Please show the units for the physical values.

[References]

Yuchen Wang, Kenji Satake, Takuto Maeda, Masanao Shinohara, Shin'ichi Sakai; A Method of Real-Time Tsunami Detection Using Ensemble Empirical Mode Decomposition. *Seismological Research Letters* 2020;; 91 (5): 2851–2861.
doi: <https://doi.org/10.1785/0220200115>

Reviewer #1

as suggested, we changed the title of the manuscript in “Volcano generated tsunami recorded in “the near-source”.

Structure of the paper was reorganized. Some of the Results were moved into the Introduction and other parts in the Methods. The body of the text, the methods and the supplementary information have been largely modified.

The Early Warning algorithm which was before in the Discussion has been moved part in the Supplementary Material and in the Methods.

Please find our reply to the points raised by the Reviewer#1 also in the annotated PDF (*Reply_to_Ref1_Comments_annotated_manuscript.pdf*).

point by point reply:

Q. ... the structure of the paper is a bit chaotic and needs to be completely reworked. Several sections appearing in the results or discussion are in the area of methods or introduction. The separation between the results and their discussion is unclear (see my suggestions in an annotated version of the manuscript).

A. Following this suggestion, we have reorganized the structure of the manuscript, posing attention in separating the material between introduction and results. An other part of the text dealing with the instruments and the detection algorithm was moved in the Supplementary material.

Q. - The literature on tsunamis generated by subaerial landslides is not well "digested", which is understandable because the authors have rather an expertise in volcanology, but this is felt in certain phrases (eg lines 263-266). This literature is certainly abundant, but it lacks a paragraph mentioning the landslide parameters that influence tsunami generation (eg from experimental and numerical studies).

A. This comment is partially correct. We did not make explicit all the parameters needed to model tsunami generated by landslide. We agree that we have been generic on this aspect on the text, because we thought that the list of parameters is given in the Table I. However, we take this suggestion and we now make it clear also in the text.

Q. - The notion of "near-field" must be defined. Many tsunami studies consider the near-field as an impacted area close to the source (typically several kilometers away), but here you are almost at the source. Your record is indeed unique, but two references should still be cited where measurements are available: Rabaul 1994 (tide-gage record of the tsunami generated by PDC: Nishimura et al 2005 In: K. Satake (ed.) Tsunamis: case studies and recent developments, Springer, 43-56) and Montserrat 2003 (dilatometer data: Mattioli et al 2007 GSA Bull 35). To come back to this idea of "near-field" I would suggest that either you keep writing "very near-field" everywhere in the manuscript, or you write "at its source" or "near-source".

A, The two mentioned references about the tsunami of Rabaul 1994 and Montserrat 2003 are cited in the manuscript.

A, We agree with the suggestion to change the definition of “very near-field” with “near-source”. This has been changed both in the title and in the text.

Q. - The discussion on existing alert systems is somewhat peremptory and involves simplifications. In any case, we cannot compare a system dedicated to ocean-wide tsunamis. For sources such as landslides and volcanic eruptions (except for atmospheric waves), local systems nested within regional systems must be used. This is, moreover, what is being done in Stromboli... but you are more aware of these steps than I am. This discussion should be redirected to the need to set up local warning systems and dedicated procedures on active volcanoes close to the coast, rather than to "criticize" the existing systems which remain essential for other types of sources.

A. It was not our intention to criticize the existing warning system, which is indeed very efficient, but only to point out the reason why we need a different approach with tsunami generated by volcanoes. This is a common and shared opinion among experts on tsunami (Bardet et al., 2003). However, we take greatly in consideration this suggestion and we have smoothed most of the sentences.

Q. - It's a shame not to have NHWAVE simulations with volumes comparable to the events of July and August 2019. This would bring real added value to your work. This is not a criticism, but a suggestion to consider.

A. We agree with the Reviewer but we prefer to leave the modeling using NHWAVE as the object of a following more dedicated paper.

Q. See also other comments on the annotated pdf attached to this report.

A. All the comments annotated in the attached pdf have been considered (see the annotated *Reply_to_Ref1_Comments_annotated_manuscript.pdf*) and the text was accordingly modified.

Referee #2

The method to localize the source of the tsunami has been changed using, as suggested, finite difference elements. The result of this new location is now described in the Method. Besides we have added a new Supplementary Video 2, to show the propagation from the source with new methodology.

All the dates in the manuscript are now in the same uniform style (DD MM YYYY).

Point by point reply

[Major comments]

Q. First of all, the structure is not well organized. Here I list up parts that may need to be revised.

A. This comment is in line with the suggestions of Reviewer#1. Many of the parts underlined by the Reviewer have been changed and/or moved in the Introduction and Supplementary Material

Q. - Lines 41–46: I do not think the Tonga eruption case needs to be explained in an independent paragraph, because this study's focus is rather on Anak Krakatau type. This part can be shortened and merged into the previous paragraph.

A. As it was suggested we shortened and merged this paragraph into the previous one.

Q.- Lines 85–105: This part does not describe results of this study and should be moved to the introduction section.

A. Lines 85-105 was moved in the Introduction

Q. - Three paragraphs in Discussion (Lines 257–261, 334–340, 392–398, and 420–428) do not fit here and most of the parts are just repeated explanations that have already appeared in earlier parts. I recommend that the authors revised the parts carefully to avoid the redundant repeating.

A. This comment is right. We have deleted some of the paragraphs and moved others in the Introduction.

Q2. Estimation of initial tsunami location and tsunami propagation speed In “Tsunami data wavefield”, the authors assumed straight ray paths at constant speeds. But clearly the tsunami wave speeds depend on the bathymetry, and the ray paths are not straight. Are the assumptions, straight ray paths and constant speeds, are truly valid? I think the ray path should travel slightly to a deeper part offshore and bent back to the shore due to the bathymetry slope. I recommend the authors perform basic tsunami simulations (for example, I guess linear long-wave simulation from a simple Gaussian-like source is enough) distributed in the same way, and compare the simulation and observed arrival times.

A. We thank the Reviewer for having point-out this method to calculate the position of the tsunami source. As suggested, we have calculated the position of the source using Gaussian-like source with a finite time difference domain model with variable tsunami celerity based on the bathymetry. The result gives the same location as for the straight ray path and a mean tsunami celerity of 39 m/s.

[Minor comments]

Q. - Lines 27–32:

Please provide references for the following parts for the two sentences, “More than 10% of all tsunamis are generated by landslides or volcano collapses with subaerial, partially submerged or submarine origins”

A. This sentence is from Titov, V. V. Hard lessons of the 2018 Indonesian Tsunamis, *Pure Appl. Geophys.*, **178**, 1121–1133., 2021).

Q. and “In recorded history landslide-generated tsunamis have triggered local wave heights and runup as large as 100 m and 500 m, respectively...”

A. This part of the sentence is now cited as Fritz, H. M., W. H. Hager, and H.-E. Minor, Lituya Bay case: rockslide impact and wave run-up, *Sci. Tsunami Hazards*, 19, 3–22, 2001.

Q. - Line 52: extent => extends

A. Done

Q. - Line 54: cost => coast

A. Done

Q. - Line 70: The connection between this and the previous paragraphs are not clear. How about starting this paragraph by “In addition to the flank collapse, the impact of the pyroclastic flow with the sea can originate tsunami with run-ups several meters high, as observed during the Montserrat 1997 and 2003 eruptions (23) and the Rabaul 1994 eruption (24)”?

A. We have now modified the text as suggested by Reviewer

Q.- Lines 76 and many other parts: The date representations are not unified; July 3rd, 3 July, or 3 July. Sometimes, 3th July.

A. All the dates have been corrected in the manuscript following the British nomenclature, that is 3 July 2019.

Q. - Line 78: which propagating => which propagates

A. Done

Q. - Line 98: “in in” => in

A. Done

Q. - Line 124: What does “involving most probably also part of the crater rim”?

A. We deleted “involving”, which was not very clear

Q. - Line 133: What do you mean by “in the second portion of the Sciara del Fuoco”?

A. This is good question. We mean in the “lower portion of ...”. We changed the text

Q. - Line 136: yellow square => yellow star

A. Done

Q. - Line 137: clear visible => clearly visible

A. Done

Q. - Line 159: during the 3 July => during the 3 July event

A. Done

Q. - Line 160: the its => its

A. Done

Q. - Line 164: concentrate granular avalanche => concentrated granular avalanche

A. Done

Q.- Line 165: almost ~200m large => almost ~200m high

A. We changed in “~200 m wide”. This is the value of the b parameter.

Q.- Sections “Tsunami data wavefield” and “Tsunami waveforms”: Please reconsider the titles of these sections. The present titles are similar, and readers may not understand the differences at first glances. Please describe what you’ve done in these sections by concise but informative titles.

A. Following this suggestion we have now named the two paragraphs as “Tsunami source time constrain” and “Tsunami wavefield characteristics”

Q.- Line 182: After 43 seconds => After 43 s

A. Done

Q. - Line 186: amplitudes and time difference => differences in amplitude and time

A. Done

Q. - Lines 204 and 210: the second tsunami on August 28 => the tsunami on August 28 (this reads as two tsunami events occurred on August 28)

A. This is correct, we changed the text as suggested

Q. - Lines 221–223: The waveform similarity between the two tsunamis (Fig. 3a,c) recorded at PLB gauge, or the positive onset and the almost identical impact velocity ($u \sim 46$ m/s) of the density current, points to the same dynamics process.

A. we changed the text as suggested

Q. - Line 305: remarkable similar => remarkably similar

A. Done

Q. - Line 319: which impacting => which impacts

A. Done

Q. - Line 329: $0.67 \pm 0.15 \times 10^5$ => $(0.67 \pm 0.15) \times 10^5$

A. we prefer to keep it without the brackets => $0.67 \pm 0.15 \times 10^5$

Q. - Line 334: the two-tsunami occurred => the two tsunamis that occurred

A. Done

Q. - Line 334: are => gives or provide

A. We changed “are” with “provide”

Q. - Line 347: the a priori => the a-priori

A. Done

Q. - Line 348: a quite difficult => quite difficult

A. Done

Q. - Line 351: wait => waiting

A. Done

Q. - Line 350: implies to => implies

A. Done

Q. - Line 355: LTA/STA method has been already attempted by a previous study for tsunami detection, for example, Wang et al. (2020).

A. Our detection algorithm was operative already in 2019, before the paper of Wang, however we now cite Wang et al., 2020 as suggested,

Q. - Line 406: previous observed => previously observed

Also, please provide references to the previous studies.

A. We corrected the text and cite Murty, T.S. (2003). Tsunami Wave Height Dependence on Landslide Volume.

Q. - Line 460: Fig. 4b => Fig. 4a

A. Done

Q. - Lines 460: what does "travel times should equals the $t_{PLB} \leq 41s$ and $t_{PDC} \leq 15 s$ time interval" mean?

A. We agree with this comment and we have deleted this part of the phrase, which was misleading

Figure, tables, and the captions

Q. - Line 694: extend => extends

A. Done

Q. - Figure 2's captions: multiple "3th July"

A. All the dates have been changed in 3 July

Q. - Figure 3a: vertical axis "distancel" => distance

A. The label of the vertical axis was corrected

Q. - Line 754: 28 August 2019 (red line) => 28 August 2019 (black line)

A. This mistake was corrected.

Q. - Figure 5a: In legend, "Ago." => "Aug."

A. corrected

Q. - Line 733: remove "always"

A. "always" was removed

Q. - Line 738: "different three different" => three different

A. corrected

Q. - Line 759: indicates that => indicates

A. corrected

Q. - Table 1: Please show the units for the physical values.

A. We have included the units for each parameter in the Table I

New cited Reference

NOAA/WDC Historical Tsunami Database at NGDC. See www.ngdc.noaa.gov/seg/hazard/tsu_db.html. (2006).

Liska R. & Wendroff B. Two-dimensional shallow water equations by composite schemes. *Int. J. Numeric. Meth. Fluids*, 30: 461 – 479 (1999).

Løvholt, F., Glimsdal, S. & Harbitz, C.B. On the landslide tsunami uncertainty and hazard. *Landslides* **17**, 2301–2315 (2020). <https://doi.org/10.1007/s10346-020-01429-z>

Murty, T. S. Tsunami Wave Height Dependence on Landslide volume, *Pure Appl. Geophys.* 160, 2147- 2153, (2002).

Okal E. A., Costas E. Synolakis C. E.,. Source discriminants for near-field tsunamis, *Geophysical Journal International*, 158, (3), 899-912, <https://doi.org/10.1111/j.1365-246X.2004.02347.x> (2004).

Bardet, JP., Synolakis, C.E., Davies, H.L., Imamura, F., Okal, E.A. Landslide Tsunamis: Recent Findings and Research Directions. In: Bardet, JP., Imamura, F., Synolakis, C.E., Okal, E.A., Davies, H.L. (eds) *Landslide Tsunamis: Recent Findings and Research Directions*. Pageoph Topical Volumes. Birkhäuser, Basel. https://doi.org/10.1007/978-3-0348-7995-8_1, (2003).

Fritz, H. M., W. H. Hager, and H.-E. Minor, Lituya Bay case: rockslide impact and wave run-up, *Sci. Tsunami Hazards*, 19, 3–22, 2001.

REVIEWERS' COMMENTS

Reviewer #1 (Remarks to the Author):

Dear authors,

In this revised version, you have addressed all my comments in an acceptable way, excepting for my comment on the lack of numerical simulation available with a similar volume as the 2019 events. I understand that you plan to work on a future paper including numerical simulations, but your answer is not what is expected here. You should rather demonstrate that at this first stage you don't need numerical models (if I'm right). Anyway, the paper is now better structured and easier to follow. I still have some minor remarks (mostly wording) that are recorded on an annotated pdf (see attached).

Congratulations for this very nice and useful contribution on these unique events.

Raphaël Paris

Reviewer #2 (Remarks to the Author):

Authors: M. Ripepe and G. Lacanna

Title: Volcano generated tsunami recorded in the near source

Manuscript #: NCOMMS-23-28596A

This includes my reviewing comments on the revised manuscript submitted by the authors, Drs. Ripepe and Lacanna, for publication from Nature Communications. First of all, I am happy to find the manuscript has been well revised and improved significantly by the authors' effort, and appreciate them for considering carefully the comments raised by reviewers including myself. Now, I recommend the publication of this article after resolving remaining problems that are all minor (some of which I have overlooked in the first review). I hope these will be helpful.

Minor comments

L. 26–36:

Better that the two paragraphs are merged into a paragraph. The first paragraph should mention the study topic, volcanic tsunamis.

L. 33: Is “In the historical record history” correct? (also typo) Maybe “In the historical record”?

L. 46: arriving almost 2 hours before the expected “normal” earthquake-generated tsunami onset
The delay time should not be “almost 2 hours”, but different from station to station depending of the distance from the source. Please consider removing the specific delay time.

L.96: the permanent tsunami gauges
“the” should be removed.

L. 97: the geophysical network

Please specify what kind of geophysical observation instruments are included in the network, and when the network was installed (maybe in Methods part around line 379).

L. 107: tsunamis waves => tsunami waves

Figure 2a: Could you add an approximated scale of this video image? for example, a note on the (approximated) elevation of the mountain’s top captured in this video is helpful enough for readers to image how large the eruptions were.

Many parts: The time unit of second is “~6 s” in some parts but “10 seconds” in other parts. They need to be unified.

L. 146: on July and August 2019 => in July and August 2019

L. 191: The tsunami on 28 August, is recorded at 10:18:20.5 UTC (65.5 s after the onset of the paroxysm) only by the PLB gauge

Could you explain here why the tsunami was not recorded by the PDC gauge?

L. 201: Differently from the first large negative wave produced by submarine slides,

Is this correct? I think that submarine slides caused both negative and positive waves, and the first polarity should depend on the direction to the station from the source (for example, see the previous paper).

Ward, S. N. (2001). Landslide tsunami. *Journal of Geophysical Research*, 106(B6), 11201–11215.
<https://doi.org/10.1029/2000jb900450>

L. 235–250: Tsunamis generated by ... and August, 2019 eruption.

I think that these three paragraphs can be merged into a paragraph, which will make easier for readers to follow the context.

L. 254: are derived

“were derived” or “have been derived” would be better to mention previous studies.

L. 291: opening to implications on our ability => implying our ability (?)

L. 347–374:

The three paragraphs sound like concluding remarks, which are independent from the subsection “tsunami detection algorithm”. How about start a new subsection at L347, for example, named as “Towards early-warning of volcanic tsunamis” (this is just a suggestion, which you may ignore if

you do not like this).

L. 357: Regardless => Regardless of

L. 391: Please check the reference number (38, 5653).

L. 392: the attenuation becomes very large also above 50s
=> the attenuation is very large even at periods above 50 s

L. 416: Tsunami source location

Thank you for revision of this part, which solidifies your argument by the simulation.

Would you please also explain that the Gaussian source is assumed here, and specify the horizontal size and peak amplitude of the Gaussian? (Since the nonlinear equations are used, the amplitude info. may affect the simulation result.)

We thank both Referees for their careful check of the manuscript and their helpful comments which have greatly improve clarity of the manuscript. Hereinafter our poi-by-point reply to their comments

Referee#1

- L.23: change “too close” in “very close”

Done it

- L. 24: “generate simulation of wave inundation”. 10 minutes is anyway not enough to start a simulation, so I would rather write that there is "a limited time to issue a warning and a forecasting."

Sentence was changed following Referee’s suggestion in "...with a limited time to issue an alert based on simulation of wave propagation and inundation"

- L. 33: “volcanic flank”.

As suggested we added also “volcanic acitivity”

- L.48: change “tsunami-genic” in “tsunamogenic”

Done it

- L.55: change “is the origin” in “is at the origin”

Done it

- L. 62: and pyroclastic flows (Auker et al. 2013 Journal of Applied Volcanology).

We did not include pyroclastic flows as suggested, but we specify now that we refer to the causalities at regional scale.

- L.85: You could also cite the seminal work of Tinti et al. 2003 (NHES) and 2006 (Bull Volc).

We have included the reference to the Tinti et al., 2003 and 2006 work.

- L.94: “huge” => “important”

“huge” was changed in “large”

- L.147. delete “most”.

Done it.

- L.151: change “This is suggesting” in “This suggests that”

Done it.

- L. 214: change “by the images” in “by visible camera”

Done it.

- L.218: change “can” in “could”

Done it.

-L. 223: “contaminated”. masked? (I don't like the use of the word contaminated here)

We think “contaminated” better describes the effect of the vertical spray jet of water on the tsunami amplitude.

- L.224: change “is reflecting” in “this reflects”

Done it.

- L.235. change “current” in “currents”

Done it.

- L.238: change “large runup” in “large wave runup”

Done it.

- L.240-241: change “the tsunami wave fenrated by coastal landslide” in “tsunami waves generated by subaerial landslide”.

Sentence was changed following Referee’s suggestion.

- L.254: change “are” in “were”

Done it.

- L.274: change “Granular” in “A granular”

Done it

- L.287-290: It's not clear what is the physical nature of the landslide, and the difference with source parameters. Please revise the sentence to make it clear.

The physical nature of the landslide and the difference with the NHWAVE modelling (lines 288-290) already described in Table I, have been made explicit.

- L.292: change “risk” in “hazard”

Done it.

- L.317. “define at the first order

We did not understand the meaning of this comment.

- L.317: change “with” in “for”

Done it.

- L.318: change “volcanic eruption” in “volcano flank collapses”

We change in “volcano flank instability”

- L.320: change “The two-tsunamis occurred on the 2019 summer provide..” in “t The two-tsunamis that occurred at Stromboli in summer 2019 provide...”

This sentence was changed as suggested.

- L.323: change “...was of only 2.59 m and fortunately had no significant...” in “...was of 2.59 m and fortunately the wave had no significant”

We changed the sentence as suggested.

- L.324-325: “which during the summer are visited by more than 5000 people every day.” put this part of the sentence in brackets.

we prefer to leave it as it is. Personally I do not like brackets.

- L. 347: change “The tsunamis generated by pyroclastic flows at Stromboli represent,...” in “he 2019 tsunamis at Stromboli represent...”

Done it.

- L.348: change "...record of a tsunami at its early stage, when is forming." In "... record of a volcanic tsunami at its early stage, when it is still forming."

This part of the sentence was changed as suggested.

- L.351-352: You could briefly remind that the solid block model overestimates the wave amplitude, whereas the granular flow model gives a good approximation.

Following this comment, we now also remind that "the granular material empirical solution better resolves the source parameters than the solid block model which overestimates the tsunami height".

- L.358: please clarify that the parameters you are referring to are "at the source".

We changed this sentence in "no physical parameters of the source are available"

- L.357: change "as first" in "as a first"

Done it.

- L.367: change "risk" in "hazard"

Done it.

- L.457: change "Granular flow" in "Granular Flow Model"

Done it

- L.481: change the Title "The Early Warning Algorithm" in "Tsunami Early Warning Algorithm"

Title was changed as suggested.

our point-by-point reply have been directly made on the Reviewer PDF document:
Reply_Comments_Referee#1.pdf.

Referee#2

- L. 26–36: Better that the two paragraphs are merged into a paragraph. The first paragraph should mention the study topic, volcanic tsunamis.
The two paragraphs were merged.
- L. 33: Is “In the hystorical record history” correct? (also typo) Maybe “In the historical record”?
“history” was deleted and the typo was corrected.
- L. 46: arriving almost 2 hours before the expected “normal” earthquake-generated tsunami onset. The delay time should not be “almost 2 hours”, but different from station to station depending of the distance from the source. Please consider removing the specific delay time.
We understand this comment, but the "2 hours" in advance described in our sentence is coming from the Kubota et al., Science paper. (ref. 10)
- L.96: the permanent tsunami gauges “the” should be removed.
“the” was removed
- L. 97: the geophysical network Please specify what kind of geophysical observation instruments are included in the network, and when the network was installed (maybe in Methods part around line 379).
*As suggested, we briefly specify at the beginning of the Methods in the “*The elastic beacons tsunami gauge system*” paragraph which instruments are used at Stromboli to monitor volcano activity*
- L. 107: tsunamis waves => tsunami waves
corrected
- Figure 2a: Could you add an approximated scale of this video image? for example, a note on the (approximated) elevation of the mountain’s top captured in this video is helpful enough for readers to image how large the eruptions were.
In Figure 2a, we overlapped the georeferenced contour map of the topography on the image of the volcano slope. This should give a precise scale of the summit crater elevation.
- Many parts: The time unit of second is “~6 s” in some parts but “10 seconds” in other parts. They need to be unified.
Time unit “second” were unified and changed in “s”
- L. 146: on July and August 2019 => in July and August 2019
“on” was changed in “in”
- L. 191: The tsunami on 28 August, is recorded at 10:18:20.5 UTC (65.5 s after the onset of the paroxysm) only by the PLB gauge. Could you explain here why the tsunami was not recorded by the PDC gauge?
The reason why the second tsunami on August 2019 was recorded only by PLB gauge is already given in the sentence (183-185) above: “.. density current severely impacted on

the elastic beacon PDC, ..., which was not operating during the tsunami on 28 August 2019".

- L. 201: Differently from the first large negative wave produced by submarine slides. Is this correct? I think that submarine slides caused both negative and positive waves, and the first polarity should depend on the direction to the station from the source (for example, see the previous paper). Ward, S. N. (2001). Landslide tsunami. Journal of Geophysical Research, 106(B6), 11201–11215. <https://doi.org/10.1029/2000jb900450>. This comment is correct. Landslide source is a dipole and when is located far underwater from the coast can generate also a positive wave. We then omit to specify the negative onset for the submarine landslide.
- L. 235–250: Tsunamis generated by ... and August, 2019 eruption. I think that these three paragraphs can be merged into a paragraph, which will make easier for readers to follow the context.
As suggested, the three paragraphs were merged together.
- L. 254: are derived. “were derived” or “have been derived” would be better to mention previous studies.
“are derived” is changed in “were derived” and previous studies have been cited.
- L. 291: opening to implications on our ability => implying our ability (?)
We changed the sentence in "with implication on ..."
- L. 347–374: The three paragraphs sound like concluding remarks, which are independent from the subsection “tsunami detection algorithm”. How about start a new subsection at L347, for example, named as “Towards early-warning of volcanic tsunamis” (this is just a suggestion, which you may ignore if you do not like this).
Thanks for this suggestion. We have included a new paragraph at the end of the “Results & Discussion” which is titled “Towards Early-Warning for Volcano-tsunami”
- L. 357: Regardless => Regardless of
Changed.
- L. 391: Please check the reference number (38, 5653).
There was a typo, “5653” was correct in “56”
- L. 392: the attenuation becomes very large also above 50 => the attenuation is very large even at periods above 50 s
we changed the sentence as suggested by the Referee.
- L. 416: Tsunami source location. Thank you for revision of this part, which solidifies your argument by the simulation. Would you please also explain that the Gaussian source is assumed here, and specify the horizontal size and peak amplitude of the Gaussian? (Since the nonlinear equations are used, the amplitude info. may affect the simulation result.)
We now specify in the Methods that the finite difference time domain method based on a nonlinear shallow-water model is using a gaussian source 1500 m large and 1 m height to locate the position of the tsunami source.